# Strong CP problem, electric dipole moment, and fate of the axion

**Gerrit Schierholz**

Deutsches Elektronen-Synchrotron DESY, Notkestr. 85, 22607 Hamburg, Germany

## Abstract

**Three hard problems! In this talk I investigate the long-distance properties of quantum chromodynamics in the presence of a topological $\theta$ term. This is done on the lattice, using the gradient flow to isolate the long-distance modes in the functional integral measure and tracing it over successive length scales. It turns out that the color fields produced by quarks and gluons are screened, and confinement is lost, for vacuum angles $|\theta| > 0$, thus providing a natural solution of the strong CP problem. This solution is compatible with recent lattice calculations of the electric dipole moment of the neutron, while it excludes the axion extension of the Standard Model.**

## 1 Introduction

QCD decribes the strong interactions remarkably well, from the smallest distances probed so far to hadronic scales, where quarks and gluons confine to hadrons. Yet it faces a problem. The theory allows for a CP-violating term $S_\theta$ in the action. In Euclidean space-time it reads

$$S = S_{\text{QCD}} + S_\theta : \quad S_\theta = i\,\theta\,Q, \quad Q = \frac{1}{32\pi^2} \int d^4x \, F_{\mu\nu}^a \tilde{F}_{\mu\nu}^a \in \mathbb{Z},$$

where $Q$ is the toplogical charge, and $\theta$ is an arbitrary phase with values $-\pi < \theta \leq \pi$. Thus, there is the possibility of new sources of CP violation, which might shed light on the baryon asymmetry of the Universe. A nonvanishing value of $\theta$ would result in an electric dipole moment $d_n$ of the neutron. The current experimental upper limit on the dipole moment is $|d_n| < 1.8 \times 10^{-13} e$ fm [1], which suggests that $\theta$ is anomalously small. This feature is referred to as the strong CP problem, which is considered as one of the major unsolved problems in the elementary particles field.

The prevailing paradigm is that QCD is in a single confinement phase for any value of $|\theta| < \pi$. The popular Peccei-Quinn solution [2] of the strong CP problem, for example, is realized by the shift symmetry $\theta \to \theta + \delta$, trading the CP violating $\theta$ term $S_\theta$ for the hitherto undetected axion.

However, it is known from the case of the massive Schwinger model [3] that a $\theta$ term may change the phase of the system. Callan, Dashen and Gross [4] have claimed that a similar phenomenon will occur in QCD. The claim is that the color fields produced by quarks and gluons will be screened by instantons for $|\theta| > 0$. 't Hooft [5] has argued that the relevant degrees of freedoom responsible for confinement are color-magnetic monopoles, realized by partial gauge fixing [6], which leaves the maximal abelian subgroup $U(1) \times U(1) \subset SU(3)$ unbroken. Quarks and gluons have color-electric charges with respect to the $U(1)$ subgroups. Confinement occurs when the monopoles condense in the vacuum, by analogy to superconductivity. This has first been verified on the lattice by Kronfeld, Laursen, Schierholz and Wiese [7]. Due to the joint presence of gluons and monopoles a rich phase structure is expected to emerge as a function of $\theta$. In Fig. 1 I show the charge lattice of quarks, gluons and monopoles for $\theta = 0$ and $\theta > 0$. For $\theta > 0$ the monopoles acquire a color-electric charge [8] proportional to $\theta$. It is then expected that the color fields of quarks and gluons will be screened by forming bound states with the monopoles, and confinement is no longer guaranteed.

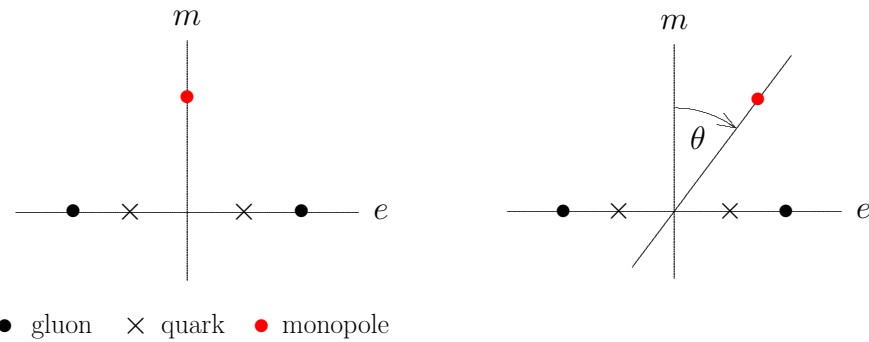

Figure 1: The color-electric – color-magnetic charge lattice for vacuum angle $\theta = 0$ and $\theta > 0$, with regard to the gauge group $U(1)$. Gluons have color-electric charge $\pm 1$, quarks have charge $\pm 1/2$, and monopoles have color-magnetic charge $\pm 1$ in Dirac units.

In this talk I will present recent lattice results [9,10] on the long-distance properties of the theory, with and without the $\theta$ term. In particular, I will show that the color fields produced by quarks and gluons are indeed screened for vacuum angles $|\theta| > 0$, thus providing a natural solution of the strong CP problem. This is compatible with recent lattice results for the electric dipole moment of the neutron. The axion extension of the Standard Model is not a valid solution.

## 2   $\theta = 0$

The core of the problem is to understand the impact of the $\theta$ term on the QCD vacuum, and on the confinement mechanism in particular. A crucial step in solving this problem is to isolate the relevant degrees of freedom. This is achieved by a renormalization group (RG) transformation, passing from the short-distance weakly coupled regime, the lattice, to the long-distance strongly coupled confinement regime. The gradient flow [11, 12] provides a powerful tool for scale setting, with no need for costly ensemble matching. It is a particular realization of the coarse-graining step of momentum space RG transformations [13–16], and as such can be used to study RG transformations directly.

The gradient flow describes the evolution of fields and physical quantities as a function of

flow time $t$. The flow of SU(3) gauge fields is defined by the diffusion equation [12]

$$\partial_t B_\mu(t,x) = D_\nu G_{\mu\nu}(t,x), \quad G_{\mu\nu} = \partial_\mu B_\nu - \partial_\nu B_\mu + [B_\mu, B_\nu], \quad D_\mu \cdot = \partial_\mu \cdot + [B_\mu, \cdot], \quad (1)$$

where $B_\mu(t,x) = B_\mu^a(t,x) T^a$, and $B_\mu(t=0,x) = A_\mu(x)$ is the original gauge field of QCD. It thus defines a sequence of gauge fields parameterized by $t$. The renormalization scale $\mu$ is set by the flow time, $\mu = 1/\sqrt{8t}$ for $t \gg 0$. The energy density at flow time $t$ is defined by $E(t,x) = 1/2 \, \mathrm{Tr} \, G_{\mu\nu}(t,x) G_{\mu\nu}(t,x)$. The expectation value of $E(t,x)$ defines a renormalized coupling

$$g_{GF}^2(\mu) = \frac{16\pi^2}{3} t^2 \langle E(t) \rangle \big|_{t=1/8\mu^2} \quad (2)$$

at flow time $t$ in the gradient flow scheme. Varying $\mu$, the coupling satisfies standard (although scheme dependent) RG equations.

We may restrict our investigations to the SU(3) Yang-Mills theory. If the strong CP problem is resolved in the Yang-Mills theory, then it is expected that it is also resolved in QCD. We use the plaquette action to generate representative ensembles of fundamental gauge fields. For any such gauge field the flow equation (1) is integrated to the requested flow time $t$. The simulations are done for $\beta = 6/g^2 = 6.0$ on $16^4$, $24^4$ and $32^4$ lattices. The lattice spacing at this value of $\beta$ is $a = 0.082(2)\,\mathrm{fm}$. Our current ensembles include 4000 configurations on the $16^4$ lattice and 5000 configurations on the $24^4$ and $32^4$ lattices each. The calculations follow [9, 10].

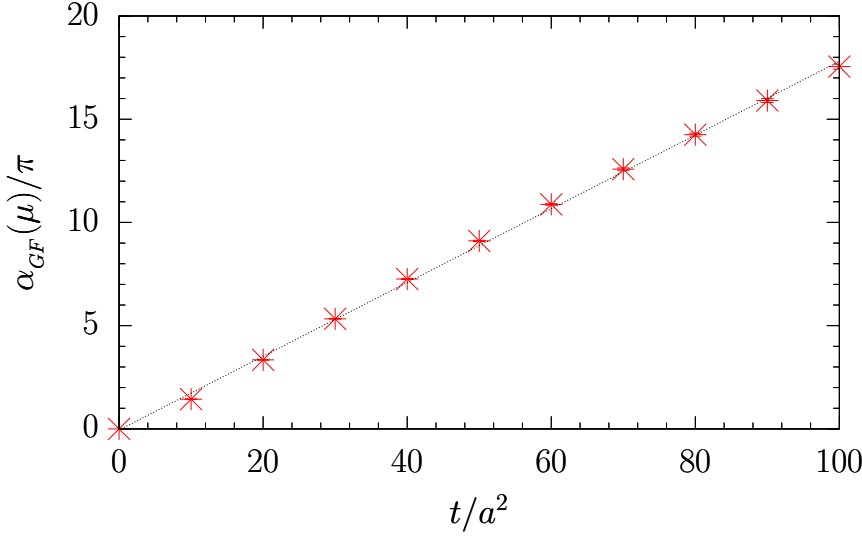

Figure 2: The gradient flow coupling $\alpha_{GF}(\mu)/\pi$ on the $32^4$ lattice as a function of $t/a^2 = 1/8a^2\mu^2$, together with a linear fit.

The long-distance properties of the theory are reflected in the parameters of the action, such as the running coupling, at infrared scales. In Fig. 2 I show the gradient flow running coupling $\alpha_{GF}(\mu) = g_{GF}^2(\mu)/4\pi$ as a function of flow time. The data continue linearly far beyond $t/a^2 = 100$, corresponding to $\mu \approx 100\,\mathrm{MeV}$, so that we may assume a strictly linear behavior of $\alpha_{GF}(\mu)$ in $t = 1/8\mu^2$. This leads to the gradient flow beta function $\partial \alpha_{GF}(\mu)/\partial \ln \mu \equiv \beta_{GF}(\alpha_{GF}) = -2\alpha_{GF}(\mu)$, which has the solution $\alpha_{GF}(\mu) = \Lambda_{GF}^2/\mu^2$ for $\mu \ll 1\,\mathrm{GeV}$ [10]. To make contact with phenomenology, it is desirable to transform the gradient flow coupling $\alpha_{GF}$ to a common scheme. A preferred scheme in the Yang-Mills theory is the $V$ scheme [17]. In this scheme $\alpha_V(\mu) = \Lambda_V^2/\mu^2$ with $\Lambda_V = 0.854 \Lambda_{GF}$.

The linear growth of $\alpha_V(\mu)$ with $1/\mu^2$, which is commonly dubbed infrared slavery, effectively describes many low-energy phenomena of the theory. So, for example, the static quark-antiquark potential, which can be described by the exchange of a single dressed gluon, $V(q) = -\frac{4}{3}\alpha_V(q)/q^2$. A popular example is the Richardson potential [18], which reproduces the spectroscopy of heavy quark systems, like charmonium and bottomonium, very well. The Fourier transformation of $V(q)$ to configuration space gives

$$V(r) = -\frac{1}{(2\pi)^3}\int d^3\mathbf{q}\, e^{i\mathbf{q}\mathbf{r}}\frac{4}{3}\frac{\alpha_V(q)}{\mathbf{q}^2+i0} \underset{r\gg 1/\Lambda_V}{=} \sigma r, \tag{3}$$

where $\sigma$, the string tension, is given by $\sigma = \frac{2}{3}\Lambda_V^2$. From a fit of $\Lambda_V$ to the data in Fig. 2 we obtain $\sqrt{\sigma} = 445(19)\,\text{MeV}$, which is exactly what we expect from Regge phenomenology.

It is interesting to compare the nonperturbative beta function with the perturbative one known up to four [19] and twenty loops [20]. In Fig. 3 the various beta functions are plotted in the $q\bar{q}$ scheme. It shows that the perturbative beta function gradually approaches the nonperturbative beta function with increasing order.

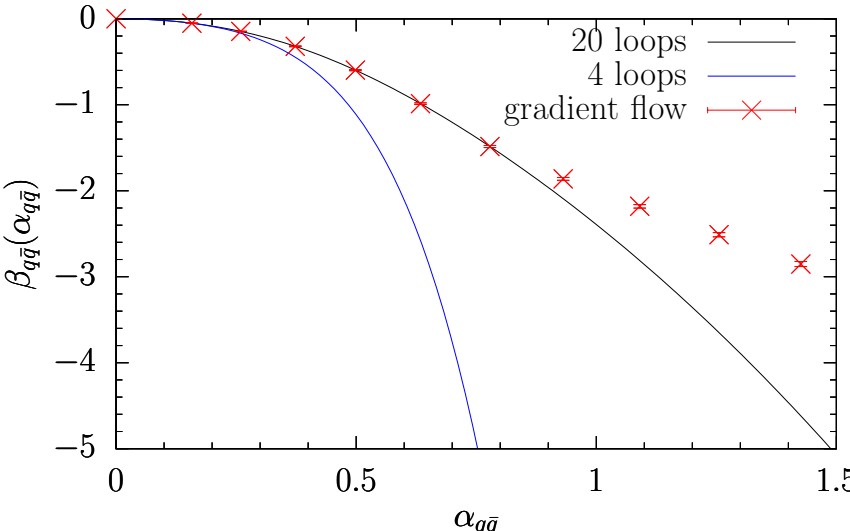

Figure 3: The beta function in the $q\bar{q}$ scheme with $\Lambda_{q\bar{q}} = 0.655\,\Lambda_V$.

In the following we will speak of confinement if and only if the running coupling extends linearly to infinity.

## 3 $\theta \neq 0$

A key point is that with increasing flow time the initial gauge field ensemble splits into effectively disconnected topological sectors of charge $Q$. This will be the case for ever smaller flow times as the lattice spacing is reduced [12]. We distinguish the topological sectors by the affix $Q$. In Fig. 4 I show the energy density in the individual topological sectors, $\langle E(Q,t)\rangle$, normalized to one for a single classical instanton.

If the general expectation is correct, and the color fields are screened for $|\theta| > 0$, we should find, in the first place, that the running coupling constant gets screened at long distances. The transformation of $\alpha_V(Q,\mu)$ from the topological sectors of charge $Q$ to the $\theta$ vacuum is

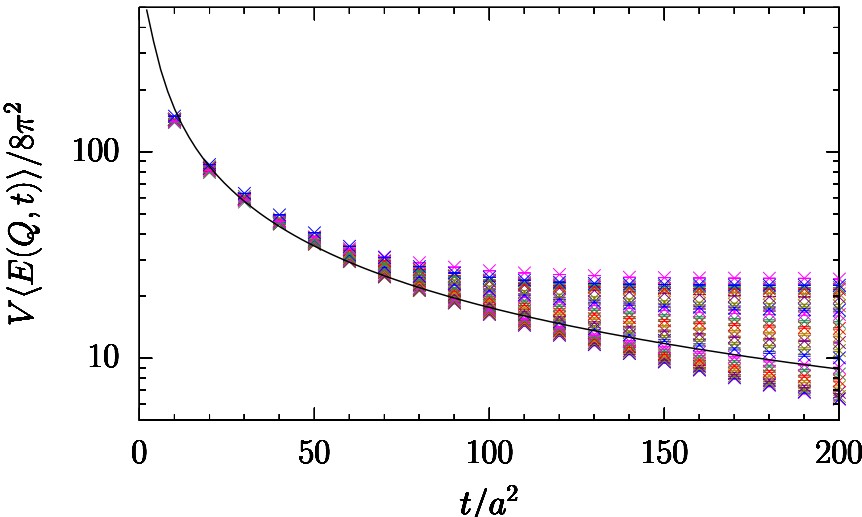

Figure 4: The action $V\langle E(Q,t)\rangle/8\pi^2$ according to $|Q|$ as a function of flow time on the $32^4$ lattice for charges ranging from $Q = 0$ (bottom) to $|Q| = 22$ (top). The solid line represents the ensemble average.

achieved by the discrete Fourier transform

$$\alpha_V(\theta,\mu) = \frac{1}{Z(\theta)}\sum_Q e^{i\theta Q}P(Q)\,\alpha_V(Q,\mu), \quad Z(\theta) = \sum_Q e^{i\theta Q}P(Q), \tag{4}$$

where $P(Q)$ is the topological charge distribution at $\theta = 0$ with $\sum_Q P(Q) = 1$. In Fig. 5 I show $\alpha_V(\theta,\mu)$ on the $16^4$ and the $32^4$ lattice. The left figure shows some finite size effects for $t/a^2 \gtrsim 50$. The 'smoothing range' $\sqrt{8t}$ should not be taken larger than the linear extent $L$ of the lattice. The effect of screening depends on the scale $\mu$, which specifies the distance

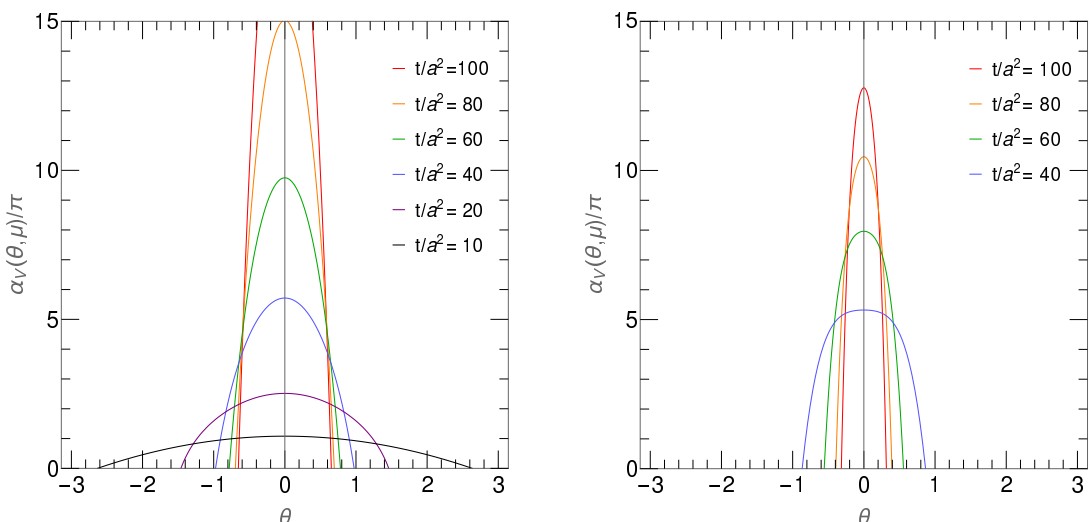

Figure 5: The running coupling $\alpha_V(\theta,\mu)$ as a function of $\theta$ on the $16^4$ (left) and the $32^4$ lattice (right) for flow times ranging from $t/a^2 = 10$ (bottom) to 100 (top). Note that $\alpha_V \simeq 0.729\,\alpha_{GF}$.

at which the charge is probed, and the angle $\theta$. At large distances ($t \to \infty$) the charge is screened for $|\theta| > 0$, while at short, perturbative distances the $\theta$ term has hardly any effect on the coupling constant. It follows that confinement is limited to $\theta = 0$.

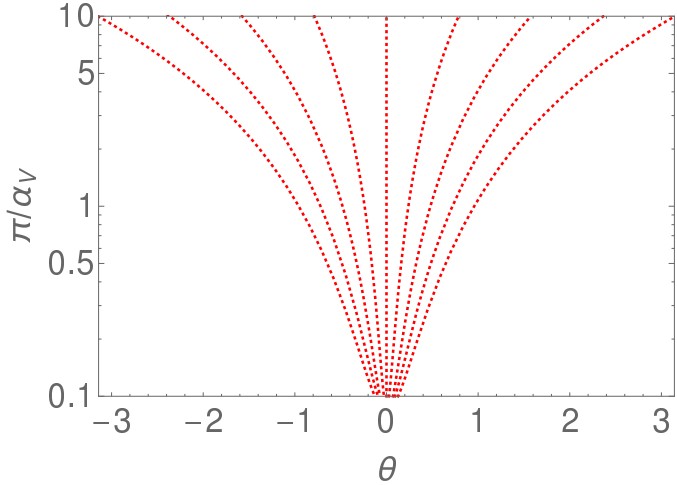

Figure 6: Flow of $\pi/\alpha_V(\theta, \mu)$ for different initial values of $\theta$ for $t$ increasing from top to bottom.

This is an important issue to understand. Let us stay with t'Hooft's model. The density of color-electric charge in the vacuum is proportional to $\theta$. Thus, the screening length will be the longer the smaller $|\theta|$ is. The result is that at asymptotic, confining distances the charge gets totally screened for $|\theta| > 0$, whereas for smaller distances, that is at larger values of $\mu$, the charge will only get totally screened once the color-electric charge density has reached a certain level, which requires increasingly larger values of $\theta$.

In [9] we have derived flow equations (see also [21]) for the running coupling $\alpha_V(\theta, \mu)$. For small values of $\theta$ and $\pi/\alpha_V$ they read $\partial(\pi/\alpha_V)/\partial \ln t \simeq -\pi/\alpha_V + D\theta^2$, $\partial \theta/\partial \ln t = -\theta/2$.

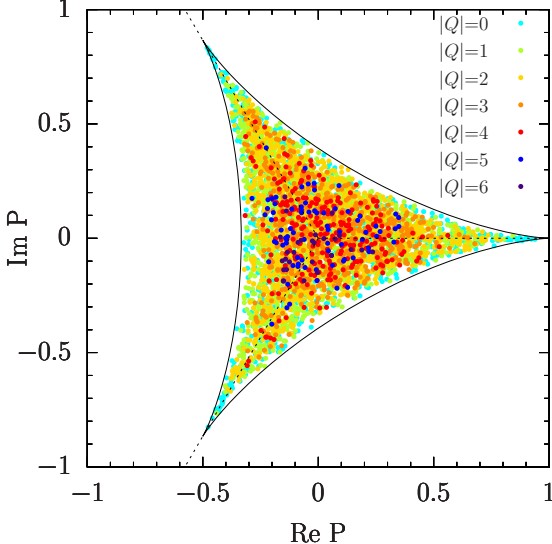

Figure 7: Scatter plot of the Polyakow loop $P$ split by topological charge $|Q|$ for $t/a^2 = 60$ on the $16^4$ lattice.

Outside this region the equations become increasingly complex. In Fig. 6 the flow equations are solved. The figure shows that any initial value of $\theta$ eventually renormalizes to zero in the infrared limit. The flow is similar to that of a scaling model of the integral quantum Hall effect [22], which has served as a model for strong CP conservation.

Let us now consider hadron observables. By nature they are RG invariant and, according to our understanding of the gradient flow, should be independent of the flow time. Two such quantities, which are easily accessible numerically and can be computed with precision, are the renormalized Polyakov loop susceptibility and the mass gap. The Polyakov loop $P$ describes the propagation of a single static quark around the periodic lattice. In Fig. 7 I show a scatter plot of $P$ at flow time $t/a^2 = 60$. We see that for small values of $|Q|$ the Polyakov loop $P$ rapidly populates the entire theoretically allowed region, while it stays small for larger values of $|Q|$. The renormalized Polyakov loop susceptibility [23] reads

$$\chi_P(\theta) = \frac{\langle |P|^2 \rangle_\theta - \langle |P| \rangle_\theta^2}{\langle |P| \rangle_\theta^2}. \tag{5}$$

It describes the connected part of the Polyakov loop correlator $\langle |P|^2 \rangle_\theta$. The transformation to the $\theta$ vacuum follows eq. (4). In Fig. 8 I show $\chi_P(\theta)$ on the $16^4$ and the $32^4$ lattice. As expected, $\chi_P(\theta)$ is independent of the flow time, and the Polyakov loop $P$ is screened for $|\theta| \gtrsim 0$.

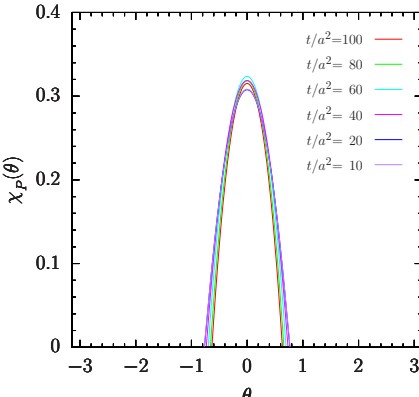
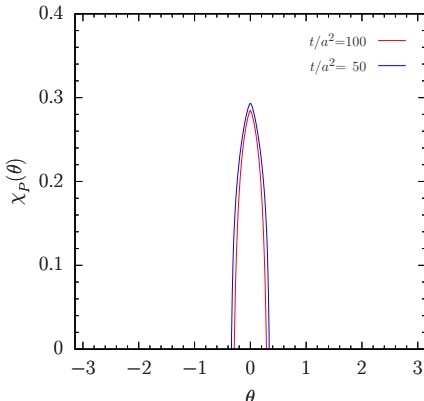

Figure 8: The Polyakov loop susceptibility as a function of $\theta$ on the $16^4$ (left) and the $32^4$ lattice (right) for flow times ranging from $t/a^2 = 10$ to 100.

The mass gap can be read off from the connected correlator of the energy density $E$. Above the vacuum, $E$ projects onto $J^{PC} = 0^{++}$ glueball states. The lowest energy state, which we denote by $m_{0^{++}}$, is called the mass gap. The inverse of the mass gap defines the correlation length, $\xi = 1/m_{0^{++}}$, which describes the length scale over that fluctuations are correlated. In the $\theta$ vacuum the glueball correlator reads

$$\langle E^2 \rangle_\theta - \langle E \rangle_\theta^2 = \frac{1}{\mathcal{N}} \sum_t \sum_{n>0} |\langle \theta | E | n \rangle|^2 \frac{e^{-m_n t} + e^{-m_n(L-t)}}{2m_n} \simeq \frac{1}{\mathcal{N}} |\langle \theta | E | 0^{++} \rangle|^2 \frac{1}{m_{0^{++}}^2}, \tag{6}$$

where $\langle E^2 \rangle_\theta = \sum_x \langle E(t,x) E(t,0) \rangle_\theta / V$ and $\mathcal{N} = L^6/16$. In eq. (6) we have assumed that the correlator is dominated by the lowest glueball state. In Fig. 9 I show $\langle E^2 \rangle_\theta - \langle E \rangle_\theta^2$ on the $24^4$ lattice. Again, the correlator turns out to be independent of the flow time, and it quickly drops to zero away from $\theta = 0$. It follows that the correlation length vanishes for $|\theta| \gtrsim 0$. This leads us to conclude that the theory has no finite mass gap for nonvanishing values of $\theta$.

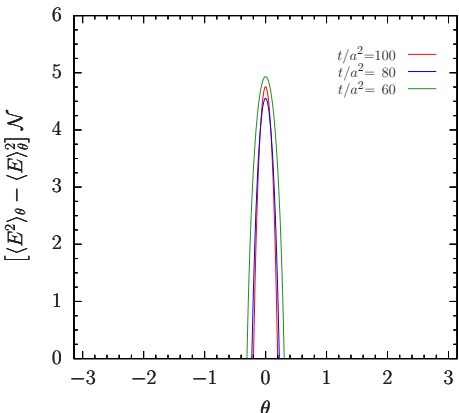

Figure 9: The connected glueball correlator on the $24^4$ lattice for various flow times.

How does this result and the result for the Polyakov loop fit together with the running coupling and the loss of confinement for $|\theta| > 0$? As I said before, the screening length decreases gradually with increasing value of $|\theta|$. For the glueball to dissipate and the Polyakov loop to be totally screened, the screening length must be smaller than the hadronic radius. On the larger volume, and for lattice spacing $a = 0.082$ fm, the Polyakov loop and the energy density appear to be totally screened for $\theta \gtrsim 0.2$. This number might decrease with increasing volume and decreasing lattice spacing. The situation here is very similar to the finite temperature phase transition.

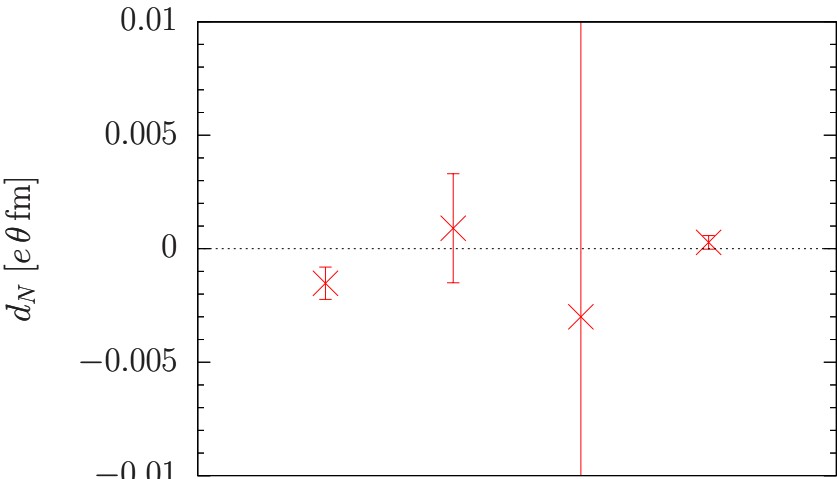

Figure 10: The electric dipolemoment of the neutron from Refs. [24], [25], [26] and [27], from left to right.

# 4 EDM

The search for an electric dipole moment (EDM) of the neutron directly from QCD constitutes a crucial test of our results. In Fig. 10 I show recent lattice results for the dipole moment of the neutron [24–27]. The results of [24–26] are at or extrapolated to the physical quark masses, while [27] refers to the SU(3) flavor symmetric point [28], where the dipole moment should be largest, while it vanishes trivially in the chiral limit. The overall result is compatible with zero. One might ask how one can find a neutron at finite, albeit small values of $\theta$. Again, this is possible as long as the screening length is larger than the nucleon radius.

In absence of a nonvanishing dipole moment, no upper limit of $|\theta|$ can be drawn from the experimental bound on $d_n$ [1].

# 5 Axion

In the Peccei-Quinn model [2] the CP violating $\theta$ term $S_\theta$ in the action is augmented by the axion interaction

$$S_\theta \to S_\theta + S_{\text{Axion}} = \int d^4x \left[ \frac{1}{2} \left( \partial_\mu \phi_a(x) \right)^2 + i \left( \theta - \frac{\phi_a(x)}{f_a} \right) q(x) \right], \quad \int d^4x \, q(x) = Q, \quad (7)$$

raising the vacuum angle $\theta$ to a dynamical variable. Under the anomalous chiral U(1) Peccei-Quinn transformation

$$U_{\text{PQ}}(1): \quad e^{i\delta Q_5} |\theta\rangle \quad \longrightarrow \quad |\theta + \delta\rangle. \quad (8)$$

It is then expected that QCD induces an effective potential $U_{\text{eff}}(\theta - \phi_a/f_a)$, having a stationary point at $\theta - \phi_a/f_a = 0$, which prompts the field redefinition $\phi_a \to \phi_a + f_a \theta$. This results in the shift

$$\theta \quad \longrightarrow \quad \frac{\phi_a(x)}{f_a}, \quad (9)$$

CP violating            CP conserving

thus effectively eliminating CP violation in the strong interaction. However, the key point is that the QCD vacuum is unstable under the Peccei-Quinn transformation (8), which thwarts the axion conjecture.

# 6 Conclusion

The gradient flow proved a powerful tool for tracing the evolution of the gauge field over successive length scales. The novel result is that color fields produced by quarks and gluons are screened for $|\theta| > 0$ by nonperturbative effects, limiting the vacuum angle to $\theta = 0$ at macroscopic distances, which rules out any strong CP violation at the hadronic level. This result does not come as a surprise. A surprise though is that the work of [3–5], for example, has been ignored for so long. Perhaps, because one did not have the right tools to attack the problem.

Screening is a gradual process, similar to the finite temperature transition. The screening length is expected to decrease with increasing value of $|\theta|$. While the color charge is screened totally at large distances, heavy quark bound states and light hadrons of finite extent will dissipate into quarks and gluons only once the screening length has become smaller than the hadronic radius.

Recent lattice results of the electric dipole moment of the neutron are found to be consistent with zero within the errorbars, in agreement with our results. However, this is not the end. The errors are rather large still, and I hope that people are not discouraged to further reduce the errors.

The nontrivial phase structure of quantum chromodynamics has far-reaching consequences for anomalous chiral transformations. In the first place that is for the axion extension of the Standard Model. The Peccei-Quinn solution of the strong CP problem is realized by the shift symmetry, $\theta \to \theta + \delta$, which is incompatible with the nonperturbative properties of the theory. Also, no light axion was found [29] in a dedicated lattice simulation of the Peccei-Quinn model. Rather, the axion mass turned out to be of the order of the $\eta'$ mass.

## Acknowledgements

Foremost, I like to thank Yoshifumi Nakamura, who generated the flowed gauge field ensembles which the calculations are based on, for long-term collaboration, and RIKEN R-CCS for computer time. Furthermore, I like to thank the organizers of this workshop for inviting me to this talk.

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
