# Peer review of "Strong CP problem, electric dipole moment, and fate of the axion"

_SciPost Physics Proceedings, doi:SciPost Phys. Proc. 6, 011 (2022)_

## Round 1 · Referee Report · Anonymous · 2022-3-27

Strengths
1- one of the most important problems of the strong interections theory is addressed
2- nice short introduction into the problem
3- the powerful tool of the gradient flow is used to address the strong CP problem
4- the string tension is computed from the lattice data for the gradient flow coupling $\alpha_{GF}$.
5- impressive comparison of the nonperturbative beta function with the perturbative one is provided to show how the perturbative beta function gradually approaches the nonperturbative one with increasing order
Weaknesses
I see none
Report
The paper is addressing one of the most important unsolved problems of the strong interections theory, the strong CP problem. SU(3) gluodynamics in lattice regularization is considered. In the Introduction a short and useful review of the problem is presented. The author uses the gradient flow to study the long-distance (small momenta) modes. The lattice configurations are produced at small enough lattice spacing to be sure that the finite cutoff effects are suppressed. The finite volume effects are kept under controle due to use of three different volumes. The gradient flow coupling $\alpha_{GF}(\mu)$ is computed numerically and $1/\mu^2$ dependence is clearly demonstrated in Fig.2. This numerical result allows to compute the string tension with value 445(19) MeV impressively close to common expectations.
In Section 3 the results for nonzero $\theta$ are presented. The lattice configurations are splitted into disconnected topological sectors after the gradient flow is applied. This allows to determine the topological charge distribution $P(Q)$ at $\theta=0$ and then compute the running coupling at nonzero $\theta$ via the discrete Fourier transformation (4). The outcome is that the the running coupling constant gets screened at long distances. While at short distances the $\theta$ term has no effect on the coupling constant. It is then concluded that confinement is limited to $\theta=0$. It is furthermore concluded via glueball correlator computations that the theory has no finite mass gap for nonzero $\theta$. The important conclusion made is that the color fields produced by quarks and gluons are screened for nonzero $\theta$ and this limits the vacuum angle to $\theta = 0$ at macroscopic distances. Thus any strong CP violation at the hadronic level is ruled out.
In my opinion the paper presents very important results on the strong CP problem obtained via numerical simulations of lattice gluodynamics and it definitely deserves to be published.

---

## Editorial Decision

published